# High-Resolution Episcopic Microscopy (HREM) in Multimodal Imaging Approaches

**DOI:** 10.3390/biomedicines9121918

**Published:** 2021-12-15

**Authors:** Katharina S. Keuenhof, Anoop Kavirayani, Susanne Reier, Stefan H. Geyer, Wolfgang J. Weninger, Andreas Walter

**Affiliations:** 1Department of Chemistry and Molecular Biology, University of Gothenburg, 405 30 Gothenburg, Sweden; katharina.keuenhof@gu.se; 2Vienna BioCenter Core Facilities GmbH, Austrian BioImaging/CMI, 1030 Vienna, Austria; anoop.kavirayani@vbcf.ac.at (A.K.); susanne.reier@vbcf.ac.at (S.R.); 3Division of Anatomy, MIC, Austrian BioImaging/CMI, Medical University of Vienna, 1090 Vienna, Austria; stefan.geyer@meduniwien.ac.at (S.H.G.); wolfgang.weninger@meduniwien.ac.at (W.J.W.); 4Zentrum für Optische Technologien, University of Applied Sciences Aalen, 73430 Aalen, Germany

**Keywords:** high-resolution episcopic microscopy, correlated multimodal imaging (CMI), correlation, bioimaging, microscopy, preclinical imaging, COMULIS, discovery pathology, histopathology, correlative morphology, phenotyping, bioimaging, mouse embryo, block face imaging, mesoscopy

## Abstract

High-resolution episcopic microscopy (HREM) is a three-dimensional (3D) episcopic imaging modality based on the acquisition of two-dimensional (2D) images from the cut surface of a block of tissue embedded in resin. Such images, acquired serially through the entire length/depth of the tissue block, are aligned and stacked for 3D reconstruction. HREM has proven to be specifically advantageous when integrated in correlative multimodal imaging (CMI) pipelines. CMI creates a composite and zoomable view of exactly the same specimen and region of interest by (sequentially) correlating two or more modalities. CMI combines complementary modalities to gain holistic structural, functional, and chemical information of the entire sample and place molecular details into their overall spatiotemporal multiscale context. HREM has an advantage over in vivo 3D imaging techniques on account of better histomorphologic resolution while simultaneously providing volume data. HREM also has certain advantages over ex vivo light microscopy modalities. The latter can provide better cellular resolution but usually covers a limited area or volume of tissue, with limited 3D structural context. HREM has predominantly filled a niche in the phenotyping of embryos and characterisation of anatomic developmental abnormalities in various species. Under the umbrella of CMI, when combined with histopathology in a mutually complementary manner, HREM could find wider application in additional nonclinical and translational areas. HREM, being a modified histology technique, could also be incorporated into specialised preclinical pathology workflows. This review will highlight HREM as a versatile imaging platform in CMI approaches and present its benefits and limitations.

## 1. Introduction

High-resolution episcopic microscopy (HREM) is a 3D episcopic imaging modality that is capable of generating volume data from whole embryos and isolated tissue specimens (biopsies). HREM is essentially an innovative adaptation of routine histology and light microscopy techniques which have been described in detail previously [1] (pp. I.2.g-1–I.2.g-9) [2,3]. As in routine histology, fixed samples are dehydrated in increasing concentrations of ethanol or methanol. Then, samples are infiltrated with and embedded in plastic methacrylate resin, which is coloured with eosin B. The polymerised resin blocks are then physically sectioned. After each section, an image is captured directly from the freshly exposed block surface using a fluorescence microscope equipped with a GFP or YFP filter-set. The preceding eosin staining provides unspecific contrast of the embedded tissues and ensures that only structures on the surface of the block are visualised. Both the optics set-up and the sample retain the same position during this procedure, and thus the images can be reconstructed to obtain three-dimensional data of the sample [1] (pp. I.2.g-1–I.2.g-9) [3,4,5,6].

Correlated multimodal imaging (CMI) represents a combinatorial approach of multiple sequential or parallel in vivo and ex vivo imaging and analysing modalities on the same tissue specimen. CMI is capable of providing structural and functional information on a tissue sample that is visualised at different lateral resolutions and penetration depths across relevant scales. This essentially involves the application of two or more complementary modalities, which in combination, provide a more informative and composite view of normal and abnormal features of the tissue specimen [7,8,9]. Prominent examples of CMI approaches that combine two modalities include:Combinations of radiology (including computed tomography (CT) and magnetic resonance imaging (MRI)) and pathology (including histopathology) provide valuable clinical diagnostic and preclinical information for better patient care or further biomedical discovery [10,11].Combinations of light microscopy and electron microscopy (EM) have been in vogue for diagnostic and research applications. In clinical and some preclinical settings, such combinations are well established in the forms of histopathology (HP) and ultrastructural pathology. In recent times, the combined application of HP, immunohistochemistry, and electron microscopy (scanning and transmission) is central to the characterisation of pulmonary lesions in fatal cases of COVID-19 [12]. In biological research, the combination of these two microscopic modalities is designated as correlative light and electron microscopy (CLEM) [13,14].

More recently, there have been CMI efforts to combine micro-magnetic resonance imaging (micro-MRI), micro-computed tomography (micro-CT), micro-Positron emission tomography (microPET), HREM, and HP, to evaluate murine tumour vasculature [15] and murine non-neoplastic vascular lesions [16] across scales. These efforts highlighted the benefits and challenges of such multimodal efforts. As expected, micro-MRI and micro-CT provided good in vivo anatomic resolution with lower tissue sensitivity, whereas micro-PET was highly sensitive but had poor spatial resolution. HREM and HP achieved the highest spatial histomorphologic resolution, but required irreversible ex vivo processing (trimming, embedding, and sectioning) [1] (pp. I.2.g-1–I.2.g-9, I.2.i-1–I.2.i-13) [15,16]. Despite some of the disadvantages, the combination of in vivo imaging followed by ex vivo processing for sequential or parallel HREM and HP can provide distinct advantages in the characterisation of sub-macroscopic and micro-anatomic morphologic defects [1] (pp. I.2.g-1–I.2.g-9, I.2.i-1–I.2.i-13) [15,16].

Due to the plethora of potentially beneficial imaging combinations, CMI has been used to tackle a variety of research questions and will continue to broaden the accessible biomedical information significantly. In this review, we highlight HREM as versatile imaging technology that helps bridge preclinical and biological imaging and integrate in vivo dynamics with ex vivo high resolution in a variety of model organisms.

### Advantages and Limitations of HREM in a Multimodal Context

The strengths and limitations of HREM are represented in Table 1 in the context of other imaging modalities that, to our knowledge, HREM has been combined with. As can be seen in Figure 1a,b, HREM assesses structural information, occupies a resolution niche that is not accessible to any other imaging modality, and connects preclinical imaging (such as micro-MRI, micro-CT, or small-animal ultrasound (US)) with biological microscopy (such as advanced fluorescence, electron, or atomic force microscopy) [7].

In imaging, penetration depth comes at the expense of lateral resolution, which restricts the scope of 3D imaging of small animals at micrometre resolution. HREM covers the mesoscopic imaging range, which refers to techniques that allow the 3D visualisation of large samples at the millimetre to centimetre scale. HREM allows the combination of a large field of view and the ability to image thick tissues of several millimetres in thickness with a high-micrometre resolution. Thus, fine structures can be analysed in the context of the overall morphology of surrounding tissues or even whole organisms. The large penetration depth of HREM is achieved by physical sectioning, which, as a downside, limits its application to dead, sacrificed samples. Since the probed volume is sectioned, images are captured from the block surface and the whole 3D sample is reconstructed virtually. There is no need for clearing the sample, as, for example, is the case in light sheet microscopy, which is best suited to optically homogeneous and relatively transparent samples. In addition, while light sheet microscopy can achieve higher resolution than HREM for similar probed volumes, the higher resolution in light sheet microscopy comes with a narrower field of view that would need to be tiled across large specimens.

## 2. State-of-the-Art

HREM is compatible with a wide range of fixatives and various contrast agents, and tissue processing for HREM does not require special chemicals except for resin embedding and contrasting with eosin. Therefore, HREM can be easily combined with almost all upstream and downstream techniques in multimodal imaging pipelines. It proved to be the method of choice to correlatively reconstruct tumour capillaries and murine vasculature of sufficient contrast and quality at micrometre resolution in selected volumes of interest (VOIs), and to visualise minute anatomical structures and volume displays of mouse, chick, and zebrafish embryos (such as heart malformations) in combination with optical coherence tomography (OCT), photoacoustic tomography (PAT), micro-US, micro-CT, and micro-MRI. HREM was shown to be compatible with contrast agent-enhanced CT and micro-MRI and with the fixation, staining, and dehydration media used after tumour and embryo removal for ex vivo micro-CT or HP. Even though HREM is destructive to the tissue, it can be combined and correlated with HP by collecting physical sections for subsequent examination.

### 2.1. HP and HREM: Mutual Complementarity

HP is well established in several non-clinical settings including translational biomedical research, model organism phenotyping, preclinical therapeutic discovery, and preclinical safety assessment. In these settings, HP is integral to study specific data generation and validation [1] (pp. I.2.i-1–I.2.i-13) [19,20,21,22,23]. HREM has been most impactful in the phenotyping of embryos and characterisation of structural developmental anomalies. HREM has also found some application in the three-dimensional visualisation of structures in tissues such as human skin and human liver, and in the characterisation of aberrant tumour vasculature and vessel wall lesions in mouse models [1] (pp. I.2.g-1–I.2.g-9) [15,16]. HREM is not typically featured in routine diagnostics, therapeutic discovery, or safety assessment [1] (pp. I.2.i-1–I.2.i-13) [19,20,21,22,23].

Founded on the same principles of histology, HREM and HP are modalities that generate valuable morphologic data at relatively low costs [1] (pp. I.2.g-1–I.2.g-9, I.2.i-1–I.2.i-13) [6,19,21,22].

HP is based on the evaluation of tissue sections on glass slides, typically by expert pathologists, followed by the acquisition of 2D images at different magnifications to represent specific features of interest observed during evaluation. HREM on the other hand is based on the acquisition of images from the cut surface of the block (rather than the section itself) through the entire thickness of the embedded tissue. The use of the lower magnification objectives incorporates a greater area of the tissue for serial imaging and 3D reconstruction but provides lower cellular detail than available in HP images.

HREM and HP focus on imaging different surfaces of the embedded tissue (block surface versus tissue section) while following the same fundamental principles of histology and light microscopy. They are closely related and mutually complementary modalities, which—when properly combined—can provide histomorphologic information with better 2D cellular detail and 3D spatial context [1] (pp. I.2.g-1–I.2.g-9) [3,4,5,6,15,16]. The similarities and differences between certain aspects of the two modalities are summarised in Table 2.

### 2.2. Multimodality HREM to Image Mouse, Chick, Quail, Frog, and Zebrafish Embryos

Developmental pathology is a well-established discipline in hospital and non-hospital practice and HP is integral to diagnosing/characterising many developmental anomalies in patients and model organisms [22,23,24,25]. Congenital defects in the embryo or foetus caused by exposure to various toxic agents including certain drugs are studied under developmental toxicologic pathology. Screening for such defects in the embryos of different rodent and non-rodent species constitutes an important component of the safety assessment of new drug candidates [20]. HP has been, and continues to be, a significant modality in the characterisation and representation of stages of normal embryonic development and certain causes of embryonic mortality [24,25].

Episcopic 3D imaging methods, and HREM in particular, have become established in life science research as valuable and impactful modalities for 3D visualisation and characterisation of normative and defective embryo anatomy. A large-scale mouse embryo phenotyping project designated as Deciphering the Mechanisms of Developmental Disorders (DMDD, https://dmdd.org.uk/, accessed 25 September 2021) is significantly based upon HREM methodology [1] (pp. I.2.g-1–I.2.g-9) [5,6].

Non-invasive in vivo imaging modalities such as micro-CT and micro-MRI are excellent tools for screening developmental phenotypes and monitoring lesions as they evolve. These modalities can provide reliable information on the location, overall structure, volume, distribution, and number of lesions, but lack sufficient resolution to distinguish specific histomorphologic lesions and cellular alterations. In some contexts, ex vivo information obtained from in vivo imaging would be incorporated into downstream pathology workflows to enable more detailed HP evaluation. Such workflows are common in clinical diagnostic and non-clinical discovery settings wherein MRI, gross pathology, and HP are combined to arrive at diagnostic conclusions or to derive relevant morphologic data [10,11,25,26]. Given these considerations, a CMI workflow as illustrated in Figure 2 would be beneficial for the systematic evaluation of embryonic/foetal anomalies in basic life science and non-clinical translational research settings.

In the following, we provide an exhaustive overview of published CMI workflows that specifically integrated HREM to diagnose and characterise embryos or foetuses.

#### 2.2.1. Visualising Gene Activity within Tissue

HREM is based on unspecific contrasting and provides highly detailed information of the overall morphology of various biologic specimens. Combining this detailed structural information with the exact localisation of specifically labelled structures, gene expression and gene product patterns is a highly beneficial tool to unravel basic developmental and pathologic mechanisms. Thus, attempts to detect specifically stained structures were undertaken from the very first beginnings of HREM.

Due to methodical constraints, the visualisation of specific signals has to be performed after wholemount staining prior to the embedding of the specimen, which is challenging for large samples and hard-to-penetrate tissues. Nevertheless, it has been achieved following wholemount in situ hybridisation, wholemount immuno-staining and lacZ staining utilising two of the multiband filter sets during HREM data generation [27]. More recently, multifluorescence HREM (MF-HREM) utilising fluorescent dyes and opaque resin has been introduced [28].

#### 2.2.2. Combining Micro-MRI and HREM for the Analysis of Murine Embryos

The production and screening of mutant embryos in large-scale phenotyping studies is expensive and time consuming. Imaging pipelines that facilitate rapid pre-selection of samples or definition of volumes of interest at lower resolution, which is then followed by detailed validation at higher resolution, could provide an efficient alternative.

Both micro-MRI and HREM have been established as routine tools for analysing phenotypes of genetically altered mouse embryos. While micro-MRI offers the advantage of being able to simultaneously scan multiple embryos at the organ and organ-system level in one run, HREM offers superior near-histological image quality. The combination of both techniques allows for identifying embryos of interest that show an ambiguous phenotype or a phenotype hinting at additional abnormalities with micro-MRI. These embryos are then subjected to HREM for further analysis [26].

This also allowed for a better management of the large amounts of data typically generated during acquisition. A combination of both modalities offered a continuous high-throughput imaging pipeline.

#### 2.2.3. From In Vivo to High Resolution Using OCT, PAT, and HREM

Chick embryos are a valuable model organism both for developmental and cancer biology. They can be used to study embryonic morphogenesis and develop experimental surgical techniques. They are also especially useful when studying tumour genesis. To establish imaging pipelines combining in vivo observations and high resolution, various modalities must be combined. OCT and PAT are techniques that have previously been successfully used in vivo [29] and in utero [30], respectively. A pipeline integrating OCT/PAT has been established for chick embryos at several developmental stages. Following these modalities, HREM then offered high-resolution insights into regions of interest [31].

Overall, this pipeline has the potential to facilitate the analysis and monitoring of tumour genesis, organogenesis, and the development of vasculature in embryos.

#### 2.2.4. CMI Pipeline to Track the Genesis of Congenital Heart Malformations

An area of interest in developmental biology is the establishment of the left-right body axis and its underlying molecular mechanisms during early development. The heart as an asymmetric organ is an important model for left-right morphogenesis, and laterality defects are the cause for many complex congenital heart defects. To track embryonic development, in vivo imaging techniques—such as small-animal US and micro-CT—are of special interest to those research questions, coupled with high-resolution analysis methods that can reveal morphological abnormalities in detail.

In the study of Desgrange et al. [32], murine embryos at different stages of development were imaged, with a special focus on the development of the heart loop. Using micro-US, embryos were visualised in vivo at E9.5, and their positions in the uterine horns of the pregnant mouse determined. This was followed by micro-CT and HREM after sacrificing the embryos at E18.5. Because of the non-invasiveness of micro-CT, it was possible to check the situs of the visceral organs within the embryo without dissection. Afterwards, organs of interest, such as the heart, were excised and prepared for visualisation with HREM. Structural abnormalities were then determined at higher resolution.

This pipeline is not restricted to analyse laterality defects and can be easily modified to focus on other organs and developmental stages.

### 2.3. Multimodality HREM to Assess Murine Vasculature

Cardiovascular pathology is a well-established discipline in hospital and non-hospital practice, and HP, in tandem with in vivo imaging methods, is integral to evaluating and diagnosing a diverse array of cardiac and vascular pathologies in patients and model organisms [19,20,23,33]. HP is essential for the characterisation of vascular lesions such as mural microthrombi, atherosclerotic plaques, mural microcalcifications, autoimmune or infectious vasculitis, vascular intimal, and medial proliferation/hyperplasia, vascular tumours, and abnormal vessels in tumours [19,23,33,34]. HREM has been used for non-routine investigative analyses of blood vessels and other structures in human liver samples and human and porcine skin samples. HREM facilitated 3D representations of the arterial and venous networks in skin samples in these efforts [1] (pp. I.2.g-1–I.2.g-9). More recently, under the auspices of CMI, HREM and HP have been applied in combination to characterise vessel wall lesions and abnormal tumour vasculature in different mouse models [15,16]. The combination of HREM and HP facilitated the definitive identification of blood vessels and distinction of non-vascular ductal structures [15].

Two published studies are compiled in the following.

#### 2.3.1. Murine Tumour Vasculature

Tumour vasculature plays a major role for tumour progression and dissemination, highlighting the importance of vascular visualisation. The application of CMI pipelines enables the depiction of multiple vascular parameters of the same tumour across scales and penetration depths.

As part of a proof-of-principle study, a novel CMI pipeline was established to characterise aberrant tumour vasculature in a murine orthotopic melanoma model. This CMI approach by Zopf et al. [15] incorporated modalities such as MRI, PET, CT, OCT, HREM, and HP (Figure 3). This novel platform explored the feasibility of combining these technologies using an extensive image processing pipeline. In this workflow, HREM offered the highest resolution of about 3 µm. Two-dimensional HREM images offered near-histologic resolution, and the structures were further resolved and confirmed by post-HREM HP. For post-HREM HP, JB-4 resin sections were collected during HREM imaging, stained, and additionally evaluated to validate the intratumoral vessels that were segmented based on the HREM images. Correlative registration was achieved between H & E-stained JB-4 sections (collected during HREM imaging) and 2D HREM images by using the morphologic landmarks of skin, subcutaneous vessels, nerve bundles, and mammary ducts. Importantly, blood vessels did not need to be perfused or selectively contrasted for 3D imaging, which is why HREM, in addition to visualising capillaries inside and outside the tumour, also allowed one to visualise capsules, blindly terminating or beginning vessels, and short collaterals inside the tumour and necrotic tissue.

#### 2.3.2. Murine Vascular Lesions

The combination of in vivo imaging with ex vivo biological microscopy enables the visualisation of both the whole organism and single structures within. With the aim of characterising vascular lesions in a genetically engineered mouse model, a novel multimodal workflow was established as described in Keuenhof et al. [16] by combining micro-MRI and micro-CT with HREM to identify and study vascular abnormalities using a targeted knockout mouse model (Figure 4). The pipeline allowed the study of the same VOI for each modality at different length scales, from an overview at the organ level (micro-MRI, micro-CT) to micrometre resolution (HREM).

For an overview at the macroscopic scale of the entire mouse vasculature and the detection of suspected vascular lesions, micro-MRI was used with a lateral resolution of about 100 × 50 µm^2^. With contrast-enhanced ex vivo micro-CT, blood vessels in the region of interest (identified as the left hindlimb by in vivo imaging) were examined in further detail at an isotropic resolution of about 15 µm which revealed a vessel suspected of blockage. Subsequently, the limb of interest was isolated en bloc including the vessel of interest, for further characterisation of the site of suspected occlusion at a resolution of 3 µm by HREM and HP. HREM suggested a narrowing of the vessel lumen and evoked suspicion of intimal hyperplasia in the affected blood vessel segment within the VOI (Figure 4). Only the inclusion of HREM in this pipeline allowed the detection of this narrowing of the vessel lumen and vascular intimal hyperplasia within the VOIs identified by CT and MRI [16].

## 3. Discussion

We strongly believe that HREM integrated into CMI workflows will be a versatile tool to help advance translational discovery. Figure 5 illustrates potential workflows for multimodal phenotyping that differ in their degree of correlation: either exactly the same sample and sections are analysed in a correlated manner using in vivo imaging, HREM and HP (workflow 1), or the sample is divided for separate analysis with HREM and HP after in vivo imaging (workflow 2), or HP and HREM are only combined after HP analysis on additional sample sets to generate complementary data (workflow 3).

To routinely implement CMI as an imaging procedure, several bottlenecks remain to be addressed: (1) it should be ensured that the handling of samples and preparation procedures are compatible across various modalities and that data quality is not compromised; (2) the same VOI must be located across imaging platforms with the help of soft- and hardware solutions; (3) software solutions that allow the correlation and handling of complex, multiscale, multimodal, and volumetric image data should be in place. The latter includes reconstruction, segmentation and visualisation. Although CMI has great potential, the above-mentioned challenges still restrict it as a commonplace workflow for biomedical researchers. Often, a single researcher’s access to various imaging modalities is limited, and there are not yet any commercial solutions, amplifying the technological challenges. Every CMI workflow needs to be set up and optimised individually, causing it to be very time consuming, expensive, and limited in throughput.

For multimodality imaging approaches that specifically integrate HREM, we identified the following challenges.

### 3.1. Unspecific Tissue Contrast in HREM and Correlation

An unfortunate downside of standard HREM is the lack of inherent specific labelling, which might require the verification of morphologies using HP or light microscopy. Although protocols for detecting specific markers exist, they are still in an experimental stage. It will be beneficial to perform complementary immunohistochemical staining or the detection of gene expression patterns by in situ hybridisation on physical HREM sections, collected during image generation.

Without the use of multimodal markers visible across imaging modalities, correlation and re-identification of VOIs is generally challenging between HREM and down- or upstream modalities due to the unspecific HREM contrast. For correlation, regions of interest, such as blood vessels, need to be highlighted specifically by manual segmentation and visual inspection, which is time consuming and substantially reduces throughput.

### 3.2. Sample Handling to Ensure Compatibility across Modalities and HREM Correlation

#### 3.2.1. Sample Preparation

HREM is compatible with all in vivo imaging technologies as long as they are used upstream of the process. To connect in vivo to ex vivo modalities, samples need to be sacrificed after in vivo imaging and immediately fixed for HREM. A rapid process is essential to preserve morphological details in their most native state. Following HREM, other downstream modalities are not always compatible with the resin embedding required for HREM.

Other modalities that can be combined with HREM often rely on contrast methods that either do not require additional preparation or are compatible with HREM. An example of the former are the OCT/PAT modalities, which rely on haemoglobin as an endogenous contrast mechanism when imaging vasculature. Even though it is advantageous to collect HREM sections during imaging for further staining and analysis, any additional image acquisition step adds to the already large amount of data collected that need handling and storing. Ideally, specific contrast mechanisms can be built into the HREM imaging process, optimising the CMI pipeline.

HREM is highly compatible with follow-up histopathological techniques (Figure 6), as mentioned in Section 2. Sufficiently thick sections cut off during HREM imaging can be collected using glass slides, and are suited for staining methods such as the commonly used H & E stain or other more specific stains that are routinely performed in HP. It is questionable whether HREM-analysed samples can be compatible with other downstream techniques such as advanced fluorescence or electron microscopy. Required fluorophores are sensitive to preparation techniques and can be quenched during HREM imaging and sample preparation. This would hinder further analysis, especially the acquisition of 3D volumes. For electron microscopy, high-quality preservation of the sample’s ultrastructure is needed, as well as additional contrast stains. The resin necessary for HREM embedding might not provide optimal conditions for other downstream modalities.

#### 3.2.2. Sample Deformations due to HREM Image Acquisition

HREM is an inherently destructive technique as during the acquisition of images of the block face, sections are continuously sliced off the sample block. The mechanical forces exerted onto the sample during trimming and processing can lead to deformations (Figure 7). This effect was shown in a study by Zopf et al. [15] in which the murine tumour vasculature was visualised. Registrations of blood vessels from different modalities were matched with each other, but submillimetre accuracy was only partially achievable with HREM. To fully match acquisitions from different registrations, in the future, it will be necessary to apply deformable registrations. There are currently no tools available that can be integrated into a fully or semi-automated workflow; manual inspection and segmentation is required. This task, however, can prove to be difficult even for experts—specifically in the light of mechanical deformations.

### 3.3. Data Handling

#### 3.3.1. Segmentation and Annotation for HREM Correlation

One important goal in HREM is to quantitatively annotate and segment the acquired volume stacks to analyse the anatomical architecture. This knowledge of complex 3D morphologies within their close-to-native context is vital for understanding the structure–function relationship. In addition, for correlative imaging, landmarks need to be defined and identified to enable the co-registration of datasets from two different imaging modalities. Either multimodal (fiducial) markers can be added that are visible across modalities but need to penetrate the sample in 3D, or—as commonly implemented—anatomical landmarks are used to enable feature-based registration. Prior to the correlation and fusion of datasets, these anatomical landmarks might need to be segmented. For HREM datasets, due to their extreme information content, sheer size, inherently low signal-to-noise ratio due to unspecific eosin staining, and the wealth of anatomical structures present, modern segmentation schemes are often not applicable. Consequently, segmentation, such as that of the vasculature, often needs to be performed largely manually and cannot be automated. In addition, many automated segmentation approaches remain inaccessible to non-specialists, and can be of limited use on data for which they were not developed or trained. Segmentation of anatomical landmarks currently presents a major bottleneck in correlating HREM data with other imaging modalities since it is time consuming and partially subjective [15]. Identifying and segmenting individual structures in HREM data is a laborious, manual task which requires the tedious inspection of thousands of sections. For instance, the authors estimate that the full reconstruction of the vasculature based on a HREM data set gained from a murine tumour [15] would take several full working days. In addition, due to the large size of one HREM dataset of several GB, segmentation in HREM is not trivial to handle in terms of computing power and visualisation [15], and requires high-end PCs and graphic cards.

Due to the challenges mentioned above, unsupervised binarisation algorithms, such as minimum error thresholding, maximum entropy thresholding, or Otsu’s single-level method, will not be enough to reliably identify and segment HREM anatomical structures. In fact, automatic segmentations generated based on thresholds or manipulations of the image histogram usually require extensive manual post-editing to achieve the desired accuracy, and solutions will need to include learning-based approaches. Despite the boost in these approaches, so far, there are hardly any automated solutions offered to tackle HREM segmentation, and the human eye currently still outperforms computers in this task in terms of precision (at the expense of time). To make matters worse, community access to such learning algorithms and high-performance clusters is very limited. In recent years, deep neural networks have been used to improve the performance of automated image segmentation. In our experience, however, HREM segmentation is currently mostly performed manually using commercial software, such as AMIRA, or freeware tools, such as ImageJ/Fiji [35]. The offered image segmentation tools in most packages are still not automated and result in laborious and time-consuming semi-automatic workflows that need to be manually adjusted and corrected due to a lack of accuracy. Ideally, a segmentation tool will be applicable across modalities and organisms and cell types since each researcher focuses on different aspects and uses a different modality to tackle specific research questions. A versatile solution that prevents new software needing to be written for each case is deep-learning segmentation algorithms. Such a generic software package will not be hard-coded but learns to adapt to the task independently from the provided data—without the need of any a priori assumptions about the morphologies.

#### 3.3.2. Large HREM Datasets

A single HREM session can produce a dataset of several GB due to its micrometre resolution and capability of visualising relatively big volumes of several mm^3^. In combination with time-resolved CMI approaches, the sheer size of the data truly enters the big data regime. Data handling and storage are currently one of the biggest bottlenecks of HREM multimodal imaging approaches. Further increases in throughput and automation will generate even more data and raise the pivotal question of how to handle these huge amounts of big data. To make matters worse, there are currently no data-handling guidelines or data retention and management plans. Existing public data archives such as EMPIAR or Cell-IDR would only be able to be used to store part of the data [36].

One approach to alleviate the situation can be data compression. Questions to be answered in the future are whether data compression can contribute significantly to saving data storage, what the strengths and limitations of such compression approaches are, and which data compression schemes will preserve the necessary resolution and information content and will allow for subsequent quantification even after compression.

### 3.4. Access to HREM Instrumentation and Expertise

To routinely implement new multimodal HREM pipelines, access to a number of expensive state-of-the art technologies is necessary, but the availability of research infrastructure and, in many cases, the necessary expertise are lacking. Numerous initiatives have been launched to provide scientists with such access to state-of-the-art technology [37]. Two of the most popular initiatives are (1) COMULIS (Correlated Multimodal Imaging in Life Sciences, https://www.comulis.eu/, accessed on 15 November 2021), a COST Action (CA17121) to promote CMI [35], and (2) Euro-BioImaging (https://www.eurobioimaging.eu/, accessed on 15 November 2021), a European research infrastructure consortium consisting of various imaging facilities, so-called Nodes, which provide open access to imaging technologies for all life science researchers.

## 4. Conclusions and Outlook

CMI and integrative data analysis are the method of choice to examine and understand the morphologic features and their mechanistic consequences in cells, tissues, and organisms in physiologic states and pathologic conditions. CMI affords the opportunity to perform such analysis on an organ or organism across all relevant spatial and temporal scales. Such carefully coordinated sequential or parallel (when needed for specialised samples and methods) implementation of in vivo and ex vivo imaging modalities is of immense benefit in generating mutually complementary and additionally validated morphologic data from the same samples and sample sets.

In the context of ex vivo light microscopic imaging, HP and HREM are mutually complementary modalities. The combined implementation of the two modalities can augment the final image data, with each modality mitigating the general limitations of the other. HREM provides the 3D spatial perspective that can be surmised, but not visually represented with routine HP. HP provides better histologic and cellular resolution which is essential for the definition and characterisation of many lesions and pathologic processes. In the biomedical research context, HREM and HP can be incorporated into CMI workflows at the end-stage (on account of irreversible tissue processing) ex vivo modalities to generate 2D and 3D histologic data to enhance and confirm the findings of in vivo imaging. In certain other settings, such as non-clinical safety assessment with well-defined and clearly established pathology workflows, HREM can be “requested as an adjunct test” along other pathology modalities. For example, in a safety assessment study with a significant developmental toxicology component, HREM with post-HREM HP could be included with routine HP and other allied modalities pertinent to the toxicologic pathology analysis.

The future outlook for CMI workflows incorporating HREM in basic life science research and academic translational discovery appears very promising [14]. For further expansion of the scope of CMI applications of HREM, joint protocols will need to be further optimised. One of the most important areas for exploration is the further refinement of immunohistochemistry and in situ hybridisation protocols for the detection of markers on JB-4 sections collected after HREM. This will provide spatial contextual molecular marker information that can be directly correlated with the 3D histology and ultrastructural data and can eliminate, or at least reduce, the need for parallel HP on paraffin sections. Given the continuous advances in machine learning and image classification for diagnostics and discovery, the possibility of refined co-registration CMI images and HREM 2D and 3D images is another area of importance for future exploration.

## Figures and Tables

**Figure 1 biomedicines-09-01918-f001:**
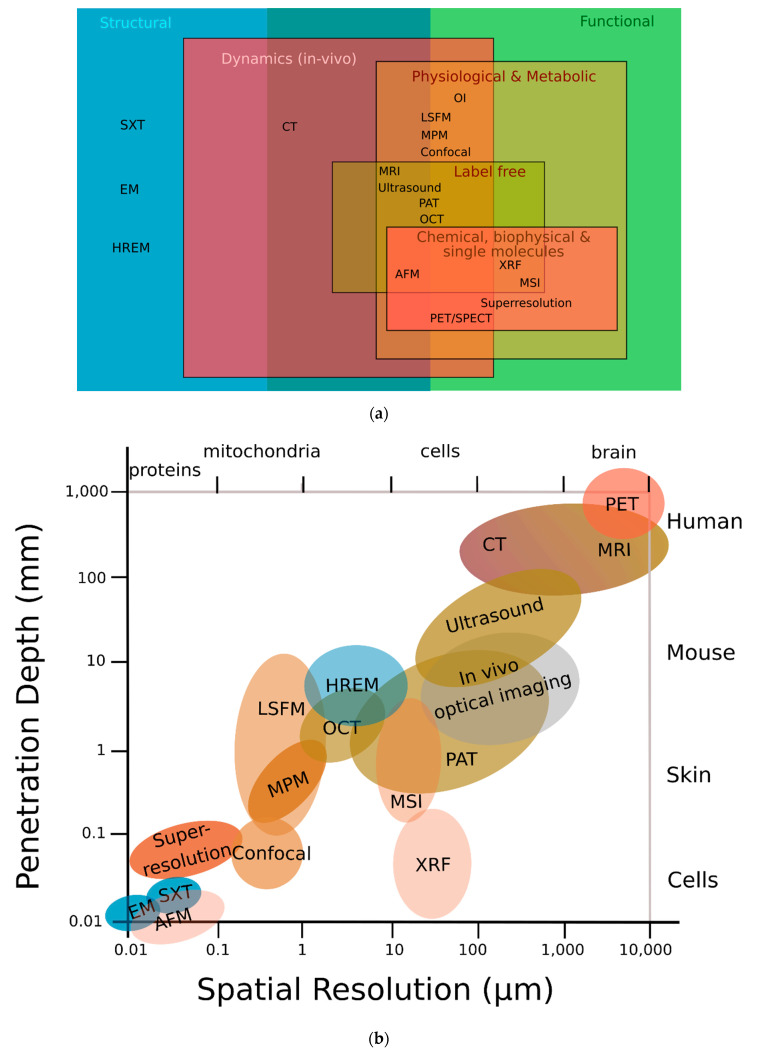
HREM can provide both high image resolution and high sample penetration depths. (**a**) Classification of different modalities according to their function; HREM offers structural information that most other modalities cannot. (**b**) HREM lies in the middle-field concerning both penetration depth and spatial resolution. Adapted with permission from [13]. (AFM) atomic force microscopy; (CT) computed tomography; (EM) electron microscopy; (HREM) high-resolution episcopic microscopy; (LSFM) light sheet fluorescence microscopy; (MPM) multiphoton microscopy; (MRI) magnetic resonance imaging; (MSI) mass spectrometry imaging; (OCT) optical coherence tomography; (OI) optical interferometry; (PAT) photoacoustic tomography; (PET) positron emission tomography; (SPECT) single-photon emission computed tomography; (SXT) soft X-ray tomography; (XRF) X-ray fluorescence.

**Figure 2 biomedicines-09-01918-f002:**
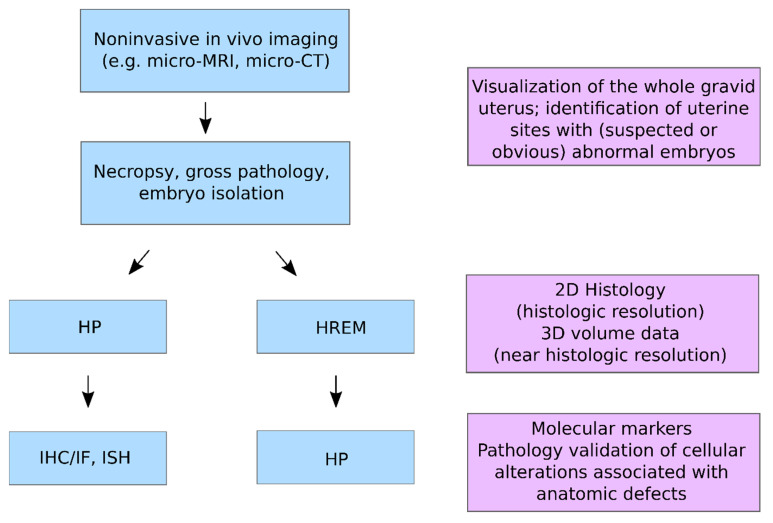
CMI workflow for embryo phenotyping in vivo imaging, HREM and HP.

**Figure 3 biomedicines-09-01918-f003:**
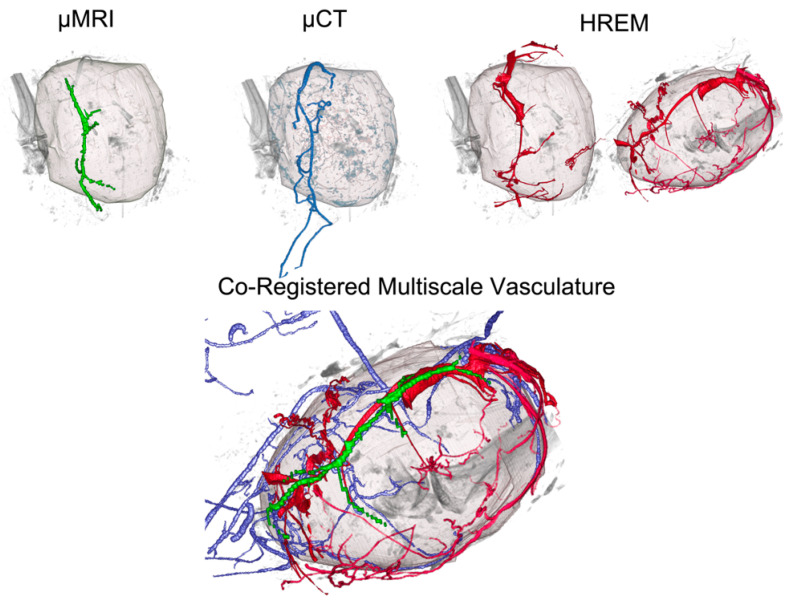
Integration of HREM data (shown in red) into a multimodal imaging pipeline of micro-MRI (green), micro-CT (blue), and HREM to reveal the vascular network of a murine tumour across scales. With a resolution of about 3 µm, HREM allowed the visualisation of blood vessels that were not detected by any of the other modalities. Reprinted from [15].

**Figure 4 biomedicines-09-01918-f004:**
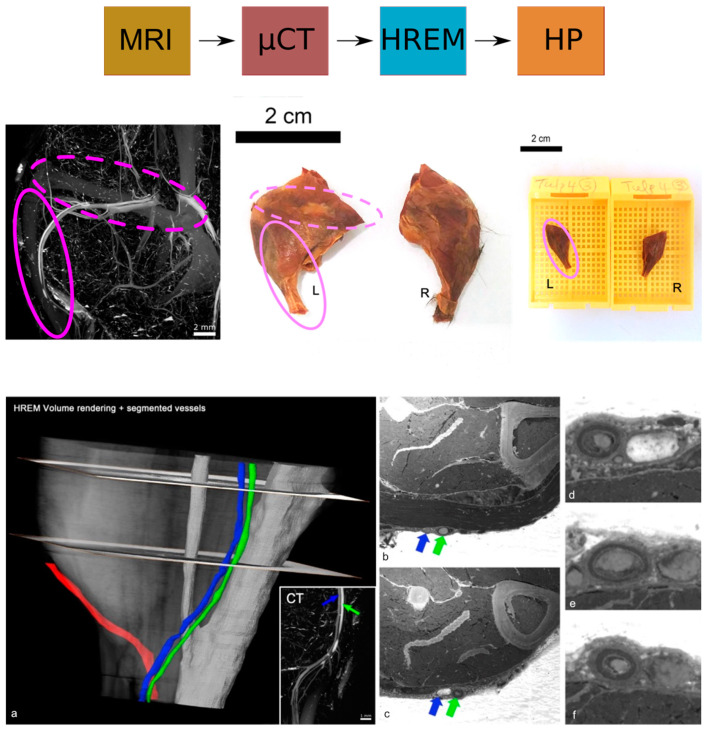
Identification of the VOI and segmented blood vessels to detect vascular lesions. (**a**) HREM and CT (inlay) volume models displaying subcutaneous blood vessels in the colours red, blue, and green; (**b**,**c**) Cross-sections of the HREM stack illustrated in (**a**); (**d**–**f**) higher magnification images of sections showing intimal hyperplasia in blood vessels. The green and blue arrow indicate the respective vessel from panel (**a**). Reprinted with permission from [16].

**Figure 5 biomedicines-09-01918-f005:**
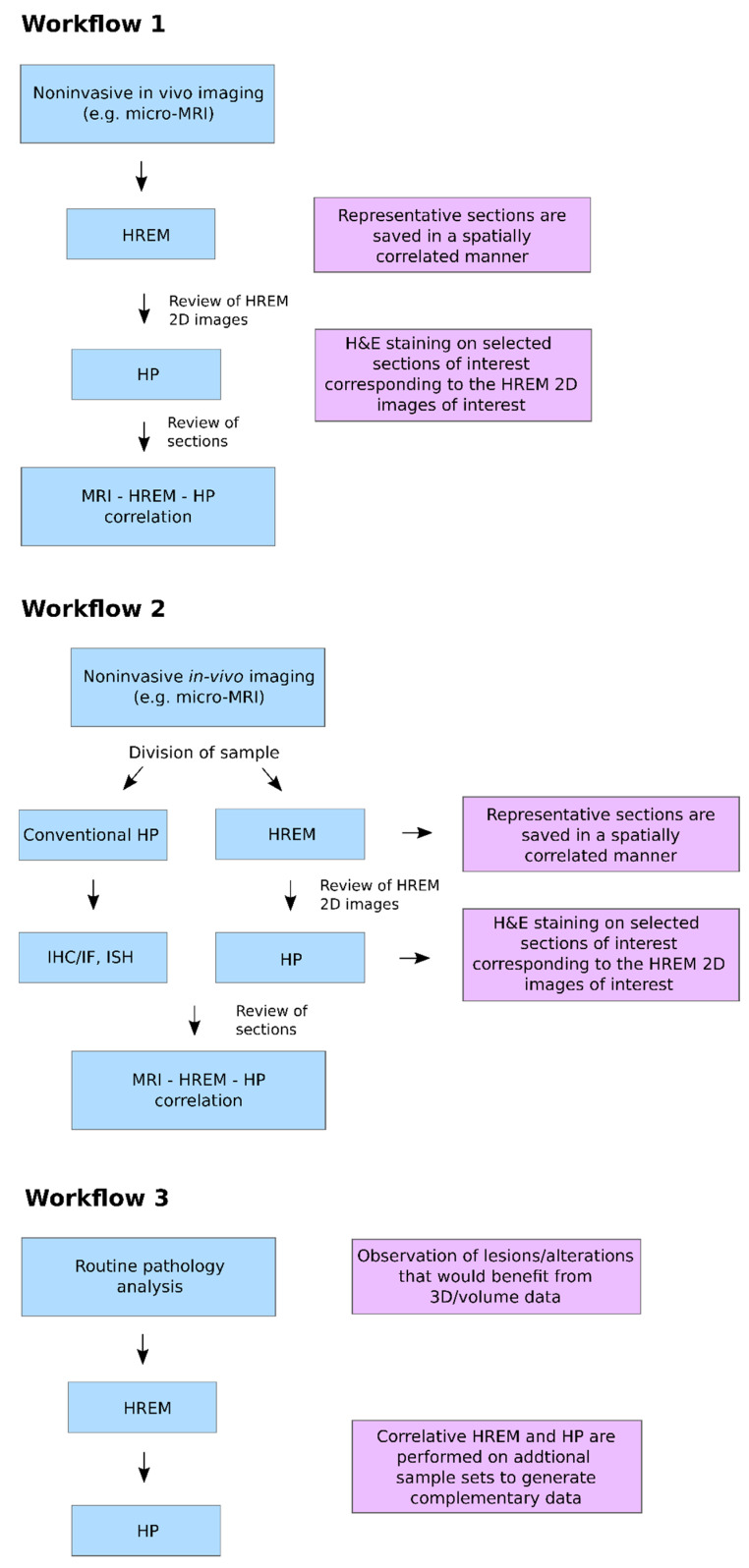
Three potential workflows for multimodal phenotyping, differing in their degree of correlation.

**Figure 6 biomedicines-09-01918-f006:**
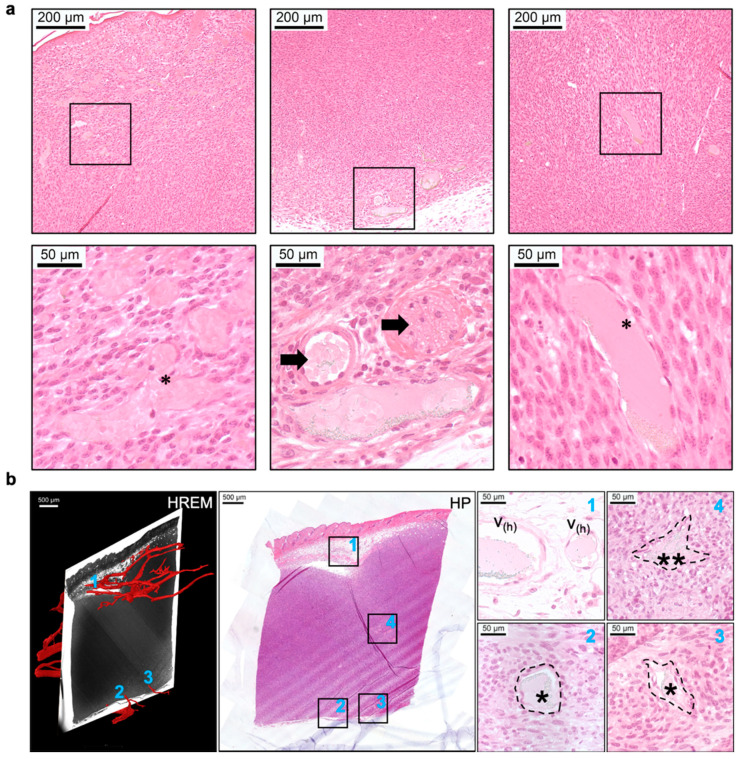
Compatibility across HREM and HP and validation of segmented blood vessels by HP. (**a**) Examples of HP sections of blood vessels, showing dilated vessels (*) and tumours (arrows), scales bars = 200 and 50 μm; (**b**) Example of co-registered images taken using HREM and HP methodologies. * and ** represent different directions of the intratumoral vessels. Scales bars = 500 and 50 μm. Reprinted from [15].

**Figure 7 biomedicines-09-01918-f007:**
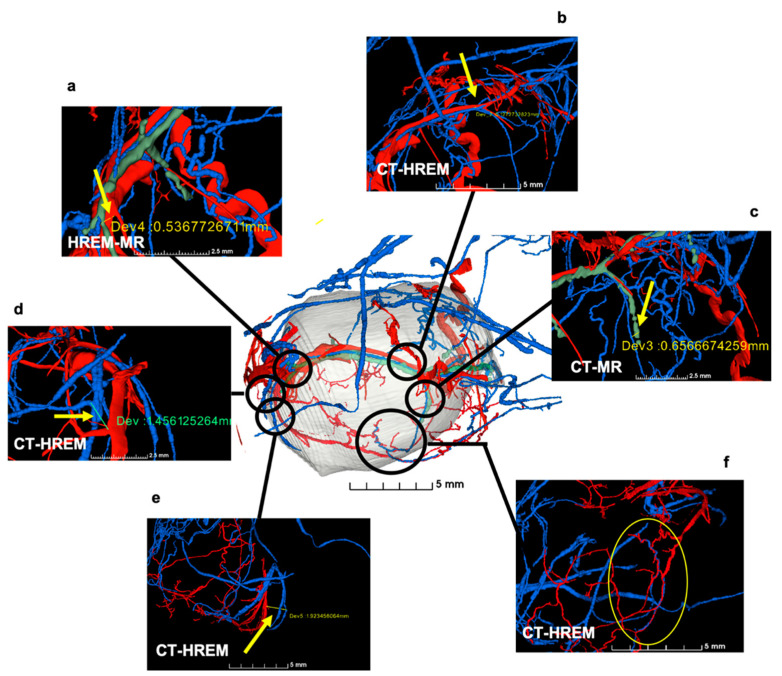
Correlation accuracy is reduced locally due to distortions induced by HREM image acquisition. Deviations in co-registrations are approximately 500 μm in (**a**), 100 μm in (**b**), 650 μm in (**c**), 1.5 mm in (**d**) and 1.9 mm in (**e**); (**f**) very close correlation of co-registered data (oval). Arrows indicate deformations in the sample caused by sectioning during HREM data acquisition. Reprinted from [15].

**Table 1 biomedicines-09-01918-t001:** Different modalities that have previously been combined in CMI studies in combination with HREM, and their imaging parameters, advantages, and limitations [1,7,17,18]. (CT) computed tomography; (HP) histopathology; (HREM) high-resolution episcopic microscopy; (MRI) magnetic resonance imaging; (OCT) optical coherence tomography; (PAT) photoacoustic tomography; (PET) positron emission tomography; (US) ultrasound.

Modality	Contrast	Penetration (mm)	Lateral Resolution (µm)	VOI	Advantages	Limitations
micro-MRI	Emitted RF signal after nuclear spin excitation	>500	≤100	whole organism	- non-ionising radiation - excellent soft tissue contrast - biochemical information (spectroscopy)	- expensive equipment - high maintenance costs
micro-US	Acoustic impedance between tissue interfaces; detection of echoes from moving particles	<150	30–800	whole organism	- high temporal and spatial resolution - portable instrumentation - cost-efficient	- limited tissue penetration - poor contrast - difficult to quantitate
PAT	Acoustic waves generated by optical absorption of tissue chromophores	~10	~40	10 × 10 mm^2^	- in vivo - penetration depth - endogenous and exogenous contrast	- resolution - speed - structural contrast
OCT	Optical scattering based on refractive index changes; motion contrast due to blood flow (OCT angiography)	~1–2	1–10 (diffraction limited)	10 × 10 mm^2^	- in vivo - fast - non-invasive - label-free - morphology - quantitative blood flow	- limited molecular information - reduced sub-cellular resolution - minimum blood flow required
micro-PET	Photon emission after positron annihilation	>500	1000–2000	whole organism	- high sensitivity - fully quantitative - broad range of applications (imaging agent dependent) - dynamic measurements	- use of radioactive agents - highly specialised equipment and staff required - high costs
micro-CT	Differential X-ray attenuation of tissues related to their density	>500	≤100	whole organism	- excellent bone imaging	- radiation dose - low soft-tissue contrast (use of contrast agents)
HREM	Light scattering based on unspecific eosin staining	Sample size up to 12 mm in thickness	>1	8 mm × 8 mm × 12 mm	- digital volumes in histologic quality at high resolution	- whole-mount contrasting of specimens - time-consuming (fixation and acquisition of 3D volume) - ex vivo, no dynamics, structural data only
HP	Light scattering (various staining methods impart colour and contrast to cellular and tissue components)	<0.1	>500	up to 1 mm^3^	- evaluation of overall tissue features at low costs - excellent cellular detail at light microscopic resolution - spatial contextual correlation of microscopic morphology	- 2D and static - detailed evaluation (especially of abnormal features in lesions) requires additional expertise

**Table 2 biomedicines-09-01918-t002:** Comparison of selected aspects of HP and HREM.

	HP _1_	HREM _2_
Fixation	Predominantly aldehyde-based fixatives (10% formaldehyde, 4% paraformaldehyde, Bouin’s fluid)	Predominantly aldehyde-based fixatives (10% formaldehyde, 4% paraformaldehyde, Bouin’s fluid)
Processing (infiltration)	Automated or manual processing Paraffin (most common) Resin, agar, gelatine, celloidin (alternative, less common)	Manual processing Resin (JB-4)
Embedding	Paraffin Resin	Resin (JB-4)
Sectioning	Manual rotary microtomy Automated microtomy	Automated microtomy
Sectioning	Single or multiple sections at specific planes of the embedded tissue for most routine diagnostic cases and discovery projects; serial sections are reserved for specialised analyses	Serial sections (at specific intervals through the entire thickness of the block)
Section thickness	1 µm to 5 µm	1 µm to 3 µm
Staining	Tissue sections are placed on glass slides and then stained	Tissues are stained during infiltration (prior to embedding)
Stains	Several histochemical stains including Hematoxylin and Eosin (H&E), Periodic Acid Schiff (PAS) and Luxol Fast Blue (LFB)	Eosin
Imaging	Light microscopy, multiple objectives and magnifications	Light microscopy, single objective and magnification (selected at the start of sectioning)
Imaging surface	Tissue section on glass slide	Cut surface of resin block
Visualisation and resolution	2D; higher histomorphologic and cellular resolution with better discernment of specific lesions _3_	2D and (virtual/reconstructed) 3D; broader spatial resolution and architectural overview with lower cellular resolution (than HP) _3_
Spatial contextual analysis of molecular (protein and nucleic acid) markers	More options for immunostaining and in situ hybridisation on paraffin embedded sections	Fewer options for immunostaining and in situ hybridisation on JB-4 resin embedded sections

Note _1_: Frozen tissues and cryosections are not discussed here. Note _2_: HREM on wholemount sections stained for reporter genes is not discussed here. Note _3_: While there are other tissue imaging modalities that provide a broader visualisation of tissue architecture or higher resolution of cellular detail, the combination of HREM with HP affords a lower-cost option that provides excellent visualisation of overall tissue architecture and histomorphologic resolution for most tissues.

## Data Availability

Not applicable.

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
