# Peer review of "High-Resolution Episcopic Microscopy (HREM) in Multimodal Imaging Approaches"

_biomedicines, 2021, doi:10.3390/biomedicines9121918_

Round 1

Reviewer 1 Report

Dear Editor,

Thanks again to have considering me for this evaluation. As you may know, evaluating a review is not exactly the same task as evaluating a scientific article. In the present case, the reviewer  has to check whether the authors discuss the field's state of the art status accuratly, obviously. However, the evaluator should also keep in mind that a review is also the place for the authors to expose their views of the field, even if he may not full agree all statements proposed in the text. Any statement diserve to be proposed and published. 

These said, here are my comments

This review is nicely written, easy to follow and I don't have any major remark.

Nevertheless I have some minor remarks that authors may think of interest for consideration.

  1. even though the authors propose a bunch of refrences describing the HREM procedure (line 38), it would be nice for readers to recapitulate the main steps in the core text.
  2. Although both concepts are evocated in the core text, it would be nice to eplicitely present the two keywords blocface imaging and mesoscopy (likely interesting to be introduced whith the concept developed lines 83-86).
  3. Position of the HREM in the whole field of imaging is presented. However, it could be interesting to hilight the positioning between microscopy and macroscopy (the keyword mesoscopy).
  4. Line 65: Although it is easy to guess that HP is for histopathology, it does not seem to be defined in the above core text.

  5. Line 92: classical HREM procedure does not retrieve any specific signal and images are ‘HE’ like images. Thus, it provides routine histological level 3D stacks (and thus 3D reconstruction). The point of clearing procedure is to allow a specific fluorescent signal hidden within the sample to be captured by the imaging system (light sheet or OPT procedure), the objective of both procedures is not absolutely the same. Nevertheless, the statement may remain valid to the condition to compare ‘fluorescent-HREM’ (the procedure is now to detect a specific fluorescent signal within the tissue, introduced line 191 – 192 in the core text) vs light sheet or OPT.

    The clearing protocols development field is still active nowadays; however, it tends to stabilize and some ‘easy to use’ robust protocols are now well accepted like iDISCO and CUBIC. Even CLARITY may be less challenging than before. Thus, the argument ‘sophisticated protocols’ for clearing to somehow disqualify these procedures is not quite strong, and could even been considered as unfair.  More accurate arguments could be: use of light sheet is limited by its width and any possible element occulting the beam. For technic like OPT, use of tomography will always be less resolutive than direct acquisition in HREM.

  6. Line 192: the capillary is the last minimal segmentation of blood vessels. Hence, it already designs a microscopic blood vessel. The semantic construction ‘microcapillary’ appear redundant.

  7. Figure 3: hardly illustrates any breakthrough in blood vessel detection in tumors brings by HREM procedure. HREM results are quite comparable to US or µMRI. µCT seems even more resolutive. This figure seems to miss its goal.

  8. Segmenting structures in HREM stacks can be challenging (discussed starting line 390). However, the authors may be to negative in their text and stating that it would take about 120 h to segmentate vasculature in a mouse's tumor seems a bit overevaluated.
  9. Size of HREM dataset could be large. However, thanks to constant increase in computers capabilities, it does not seems to be a significant issue, still.

Author Response

Thank you for the constructive comments in regards to our manuscript. We have amended the suggested changes and provide point-by-point replies in green below.

1. even though the authors propose a bunch of refrences describing the HREM procedure (line 38), it would be nice for readers to recapitulate the main steps in the core text.

We have inserted a short description of the basic HREM-steps (line 40-47).

2. Although both concepts are evocated in the core text, it would be nice to eplicitely present the two keywords blocface imaging and mesoscopy (likely interesting to be introduced whith the concept developed lines 83-86).

We have edited the corresponding paragraph accordingly (line 95-101).

3. Position of the HREM in the whole field of imaging is presented. However, it could be interesting to hilight the positioning between microscopy and macroscopy (the keyword mesoscopy).

We edited the corresponding paragraph accordingly (line 95-101).

4. Line 65: Although it is easy to guess that HP is for histopathology, it does not seem to be defined in the above core text.

We have now defined it in the text above the first use of the acronym.

5. Line 92: classical HREM procedure does not retrieve any specific signal and images are ‘HE’ like images. Thus, it provides routine histological level 3D stacks (and thus 3D reconstruction). The point of clearing procedure is to allow a specific fluorescent signal hidden within the sample to be captured by the imaging system (light sheet or OPT procedure), the objective of both procedures is not absolutely the same. Nevertheless, the statement may remain valid to the condition to compare ‘fluorescent-HREM’ (the procedure is now to detect a specific fluorescent signal within the tissue, introduced line 191 – 192 in the core text) vs light sheet or OPT.

The clearing protocols development field is still active nowadays; however, it tends to stabilize and some ‘easy to use’ robust protocols are now well accepted like iDISCO and CUBIC. Even CLARITY may be less challenging than before. Thus, the argument ‘sophisticated protocols’ for clearing to somehow disqualify these procedures is not quite strong, and could even been considered as unfair.  More accurate arguments could be: use of light sheet is limited by its width and any possible element occulting the beam. For technic like OPT, use of tomography will always be less resolutive than direct acquisition in HREM.

Thanks for pointing this out! We modified the comparison based on your arguments and removed sophisticated protocols.

6. Line 192: the capillary is the last minimal segmentation of blood vessels. Hence, it already designs a microscopic blood vessel. The semantic construction ‘microcapillary’ appear redundant.

We very much agree. Thanks, we changed it accordingly.

7. Figure 3: hardly illustrates any breakthrough in blood vessel detection in tumors brings by HREM procedure. HREM results are quite comparable to US or µMRI. µCT seems even more resolutive. This figure seems to miss its goal.

Thanks for the remark. We would like to point out that the HREM resolution (as indicated by scale bars within the subfigures) is about 6 times higher than that of microCT. However, we agree that this figure fails to illustrate this clearly for the reader. We therefore suggest replacing the figure by the rendered tumor which shows the vasculature visualized by microCT, microMRI and HREM by colors. We think that this figure illustrates the resolutive power of HREM better since HREM allowed the authors to visualize blood vessels that were not detected by any of the other modalities. We have updated the caption and figure accordingly.

8. Segmenting structures in HREM stacks can be challenging (discussed starting line 390). However, the authors may be to negative in their text and stating that it would take about 120 h to segmentate vasculature in a mouse's tumor seems a bit overevaluated.

Segmentation of HREM-data can be very time-consuming. This is, on one hand, caused by the sheer amount of morphologic data. On the other hand, the lack of specific markers hampers automatic detection of borders between different structures. For instance, segmentation of blood vessels can be speed up by using (semi-)automated segmentation tools, given that blood vessels are free from blood. However, this was not the case in the tumor mentioned in the manuscript. Moreover, due to its size, the tumor was divided into three parts, which were embedded and imaged separately. Thus, the 120 hours refer to the microvasculature imaged in three HREM-data sets, each consisting of thousands of images.

Still, we agree, that our statement concerning segmentation-time might be misleading and too negative. We therefore edited that part of the manuscript (line 416, 422-424).

9. Size of HREM dataset could be large. However, thanks to constant increase in computers capabilities, it does not seems to be a significant issue, still.

We edited the corresponding text passage (line 426-427).

Reviewer 2 Report

Manuscript ID: biomedicines-1467340 -

In this review the authors highlight the importance and the advantages of including HREM technique in a multimodal approach for preclinical and biological research analysis.

The technique is very interesting for the analysis of large sample (i.e. embryos or tissue) because of good resolution, both in depth and lateral resolution, and for the possibility to be included in CMI pipeline for multimodal analysis. The idea to combine more imaging modalities, that provide multiple various information for a more comprehensive view of the same sample is certainly an helpful approach in routine biomedical analysis for the diagnosis of different pathologies.

I think that the review paper is well written. It explains the characteristics of the HREM also in comparison with the other techniques  to visualize large biological specimens and to create 3D volume rendered model.  They propose also workflows in which HREM could be combined for a multimodal phenotyping. Examples of HREM application in combination with other imaging techniques are provided  in analysis of embryos and vasculature apparatus..

Moreover it is important to underline that the development of algorithms and software able to analysis and integrate all the data obtained by the same sample are essential to attain a pervasive, fast and reliable characterization of the analyzed samples. Only when these advances will be accomplished, a multimodal approach could be really employed in biomedical analysis.

Hence I deem that the review can be published after minor revisions:

-           The HP acronymous has not been defined. Could be helpful for the reader to explain this acronymous and discuss better what they mean and include in HP techniques.

-           In the legend the acronymous of the microscopy techniques reported of Fig1 and table 1 should be specified.

-           Pg 5: replace “table 1” with “table 2”.

-           Pg4 line 105, they state that HREM does not require special chemical for staining, but in the table 1 and table 2 it is reported that eosin staining is needed. Could you clarify this point?

Author Response

Thank you for input on our manuscript. We have amended the suggested changes and provide point-by-point replies in red below.

  1. The HP acronymous has not been defined. Could be helpful for the reader to explain this acronymous and discuss better what they mean and include in HP techniques.

The acronym has been defined accordingly in the text now.

  1. In the legend the acronymous of the microscopy techniques reported of Fig1 and table 1 should be specified.

Definitions for all acronyms in the respective figure/table have been added.

  1. Pg 5: replace “table 1” with “table 2”.

Thank you for pointing that out, it is now corrected.

  1. Pg4 line 105, they state that HREM does not require special chemical for staining, but in the table 1 and table 2 it is reported that eosin staining is needed. Could you clarify this point?

We are sorry for the inaccurate wording and changed the relevant part of the manuscript (line 116-119).